# ALIGN YOUR INTENTS: OFFLINE IMITATION LEARNING VIA OPTIMAL TRANSPORT

## ABSTRACT

Offline Reinforcement Learning (RL) addresses the problem of sequential decision-making by learning optimal policy through pre-collected data, without interacting with the environment. As yet, it has remained somewhat impractical, because one rarely knows the reward explicitly and it is hard to distill it retrospectively. Here, we show that an imitating agent can still learn the desired behavior merely from observing the expert, despite the absence of explicit rewards or action labels. In our method, AILOT (Aligned Imitation Learning via Optimal Transport), we involve special representation of states in a form of intents that incorporate pairwise spatial distances within the data. Given such representations, we define intrinsic reward function via optimal transport distance between the expert's and the agent's trajectories. We report that AILOT outperforms state-of-the art offline imitation learning algorithms on D4RL benchmarks and improves the performance of other offline RL algorithms by dense reward relabelling in the sparse-reward tasks.

## 1 INTRODUCTION

Over the past years, offline learning has remained both the most logical and the most ambitious avenue for the development of RL. On the one hand, there is an ever-growing reservoir of sequential data, such as videos, becoming available for training the decision-making of RL agents. On the other hand, these immense data remain largely unlabeled and unstructured for gaining any valuable guidance in the form of a tractable objective for learning the underlying policy, stimulating the development of unsupervised and self-supervised methods (Singh et al., 2020; Li et al., 2023; Sinha et al., 2022; Yu et al., 2022; Eysenbach et al., 2018; Park et al., 2023b).

Following the success of language-based foundational models, the offline RL community have also begun to leverage the weakly-labeled data. However, incorporating such offline data into RL frameworks efficiently remains a challenge (Levine et al., 2020)). Several factors, such as distribution shift, slow convergence when the labels are missing, and the absence of known rewards or task-specific objectives, hinder the development of offline RL (Kumar et al., 2021; Fujimoto et al., 2019).

A potential remedy to these issues is found in *Imitation Learning* (IL), where the explicit reward function is not needed. Instead, an imitating agent is trained to replicate the behavior of the expert. Behavior cloning (BC) (Ross & Bagnell, 2010) frames the IL problem akin to classical supervised learning, seeking to maximize the likelihood of the actions provided under the learner's policy. Despite working well in simple environments, BC is prone to accumulating errors in states, coming from different distributions other than that of the expert. Other notable methods include Inverse RL (Torabi et al., 2018; Zolna et al., 2020) and action pseudo-labeling (Kumar et al., 2020), however they are rarely used in practice due to the introduced overhead. Distribution matching is another promising IL paradigm, where the approaches such as DIstribution Correction Estimation ("DICE"), including SMODICE (Ma et al., 2022b), attempt to match the state-occupancy measures between the imitator's and the expert's policies. However, these methods posses several limitations: 1) they require a non-zero overlap between the supports of the agent and the expert and 2) they mostly use KL-divergence (or, generally, $f$-divergences), which ignores the underlying geometry of the spaces where the distributions are defined.

Another line of development is associated with *Computational Optimal Transport*, a domain that has garnered significant popularity for addressing diverse machine learning tasks (Peyré et al., 2019). Its applications span from domain adaptation to generative modeling (Salimans et al., 2018; Rout

et al., 2021; Korotin et al., 2022), including weakly-supervised and unsupervised methods (Bespalov et al., 2020; 2022). Optimal transport theory was proposed to alleviate the necessity for manual reward engineering (Luo et al., 2023) via establishing an optimal coupling between a few high-quality expert demonstrations and the trajectories of the learning agent, which proved to be useful for Imitation Learning tasks (Haldar et al., 2022; 2023). However, performing OT matching in the high-dimensional unstructured space is difficult, thus limiting the approach to low-dimensional tasks.

The absence of reward labels is not the sole challenge; obtaining access to the expert actions can also be problematic. When no rewards or action labels accompany the abundant demonstrations, one potential remedy is to create a simulator (Krylov et al., 2020; Saboo et al., 2021) that enables access to both actions and rewards. However, adopting such an approach entails substantial additional work. Here, we eliminate the requirement for knowing the rewards or the action labels of the expert altogether. We propose to map the initial state space to the *space of intentions*, where there is *some* high-level semantic imaginary goal that the agent needs to achieve to imitate the expert. Aligning intents of the agent with those of the expert via Optimal Transport represents a new pathway for training efficient offline RL models.

Our contribution are as follows:

- A new intrinsic dense reward relabelling algorithm that 'comprehends' the demonstrated dynamics in the environment, on top of which any offline RL method can be used.
- We *outperform the state-of-the-art* models in the majority of Offline RL benchmarks. We do so even without knowing the expert's action labels and the ground truth rewards.
- We report extensive comparison with previous works. We show that our approach enables custom imitation even if the agent's data are a mix of random policies.

## 2 RELATED WORKS

In this study, we broaden the application of offline Reinforcement Learning (RL) to datasets that lack both the rewards and the action labels. Despite all the research effort on learning the intrinsic rewards for RL, most works assume either online RL setting (Brown & Niekum, 2019; Yu et al., 2020; Ibarz et al., 2018) or the unrealistic setup of possessing some annotated prior data. Moreover, the problem of learning and exploration in sparse-reward environments can be deemed as solved for online RL (Li et al., 2023; Eysenbach et al., 2018; Lee et al., 2019); whereas, only a few works exist that consider the offline goal-conditioned setting with reward-free data (Park et al., 2023a; Zheng et al., 2023; Wang et al., 2023a). The goal of our work is to extract guidance from the expert by forcing optimal alignment of information-rich representations of imaginary goals between the expert and the agent in a shared distance-aware latent space.

We build upon the approach introduced in OTR (Luo et al., 2023), extending it by finding a representative distance-preserving isometry to the shared latent space and forcing alignment via optimal transport in this metric-aware space. In OTR, the authors estimate the optimal coupling between expert and agent trajectories, which enables rewards relabeling for agent's state transitions by computed Wasserstein distance. However, performing optimal transportation matching in this way is sensitive to underlying cost function (e.g Cosine or Euclidean) and final results can vary drastically. Choosing right distance function requires additional knowledge of the environment and extensive search, limiting application to simple tasks.

An advantage of our method is in its ability to perform distribution alignment in the space, which captures temporally structured dependencies in the provided datasets, which is completely ignored in previous works. For example, a lot of methods focus on KL divergence, which is agnostic to the distance metric Such temporal structure induces spatial closeness between temporally simil-iar states.Additionally, the method seamlessly integrates with various downstream RL algorithms, providing flexibility in selecting the most suitable training approach.

Another recent method called Calibrated Latent gUidancE (CLUE) (Liu et al., 2023), introduces a parallel approach for deriving intrinsic rewards. In this study, the authors employ a conditional variational auto-encoder trained on both expert and agent transitions and compute the euclidean distance between the collapsed expert embedding and the agent trajectory. The expert embedding might not collapse into a single point in a multimodal expert dataset, requiring clustering to handle

different skills. Unlike the CLUE method, our approach doesn't need state-action paired labeled data, eliminating the need for action annotations.

## 3 PRELIMINARIES

### 3.1 PROBLEM FORMULATION

A standard Reinforcement Learning problem is defined as a Markov Decision Process (MDP) with tuple $\mathcal{M} = (\mathcal{S}, \mathcal{A}, p, r, \rho_0, \gamma)$, where $\mathcal{S} \subset \mathbb{R}^n$ is the state space, $\mathcal{A} \subset \mathbb{R}^m$ is the action space, $p$ is a function describing transition dynamics in the environment $p \colon \mathcal{S} \times \mathcal{A} \to \mathcal{P}(\mathcal{S})$, $r \colon \mathcal{S} \times \mathcal{A} \to \mathbb{R}$ is a predefined extrinsic reward function, $\rho_0$ is an initial state distribution and $\gamma \in (0, 1]$ is the discount factor. The objective is to learn policy $\pi_\theta(a|s) \colon \mathcal{S} \to \mathcal{P}(\mathcal{A})$ that maximizes discounted cumulative return $\mathbb{E}[\sum_{t=0}^{\infty} \gamma^t r(s_t, a_t)]$. In contrast, offline RL assumes access to a pre-collected static dataset of transitions $\mathcal{D} = \{(s_i, a_i, s_i', r_i)\}_{i=1}^n$ while prohibiting additional interaction with the environment. In this study, we assume access to a dataset of transitions without reward labels $\mathcal{D}_a = \{(s_i, a_i, s_i')\}_{i=1}^n$, collected from $\mathcal{M}$, and a limited number of ground truth demonstrations from the expert policy without reward and action labels $\mathcal{D}_e = \{(s_i, s_i')\}_{i=1}^m$. These demonstrations are collected from the same MDP ($\mathcal{M}$) and we assume that several trajectories from $\mathcal{D}_e$ have high cumulative return. The primary objective is to determine a policy that closely emulates the behavior of the expert and thereby maximises the cumulative return.

### 3.2 REWARD RELABELLING THROUGH OPTIMAL TRANSPORT

Optimal Transport provides a reasonable and efficient way of comparing two probability measures supported on high-dimensional measure spaces. Given a defined cost function $c(\cdot, \cdot)$ over $\mathcal{S} \times \mathcal{S}$, the Wasserstein distance between the agent trajectory $\tau^a = \{s_1^a, s_2^a, \cdots, s_T^a\} \subset \mathcal{D}_a$ and the expert trajectory $\tau^e = \{s_1^e, s_2^e, \cdots, s_T^e\} \subset \mathcal{D}_e$ is defined as:

$$W(\tau^a, \tau^e) = \min_{P \in \mathbb{R}^{T \times T}} \sum_{i=1}^{T} \sum_{j=1}^{T} c(s_i^a, s_j^e) P_{ij}, \tag{1}$$

satisfying the marginal constraints, with $p^a, p^e$ being the state occupancy of the agent data and the expert dataset respectively:

$$\sum_{i=1}^{T} P_{ij} = p^e(s_j); \quad \sum_{j=1}^{T} P_{ij} = p^a(s_i). \tag{2}$$

Then, a proxy reward function, given the optimal transport plan $P^*(\tau_a, \tau_e)$ to transform the agent state $s_i^a$ into the closest state of the expert, is:

$$r_i = -\sum_{j=1}^{T} c(s_i^a, s_j^e) P_{ij}^*(\tau_a, \tau_e). \tag{3}$$

In this work, instead of computing the optimal transport between the initial states from the datasets, we propose to extract intentions of the expert in the form of an isometry map to the latent space, which enables a more accurate modelling of the distribution of the similar states (based on temporal distances between them).

### 3.3 PRETRAINING OF DISTANCE PRESERVING REPRESENTATION

Using pre-collected large unlabeled datasets in the form of prior data enables learning diversified behaviors in the form of skills or general representations. Even in the absence of the ground truth actions or rewards, the policies can still acquire valuable insights about the dynamics in the environment. Several works tried to define valuable objective for learning from the unstructured data, including offline GCRL (Eysenbach et al., 2022; Park et al., 2023a) and skill discovery (Eysenbach et al., 2018; Sharma et al., 2019). Finding general mapping $\psi \colon \mathcal{S} \to \mathbb{R}^d$ which is a lossless compression of a state was addressed in several formulations including *successor features* (SFs) (Dayan, 1993; Barreto et al., 2017; Borsa et al., 2018) and Forward-Backward (FB) representations

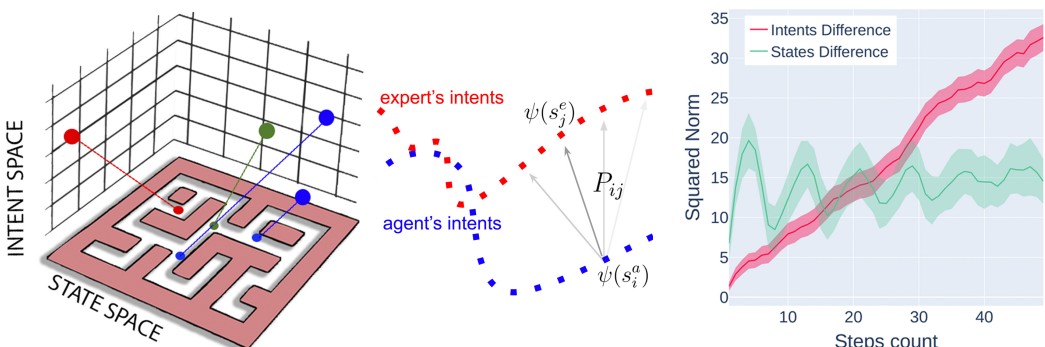

Figure 1: AILOT: Aligned Imitation Learning via Optimal Transport. Principal diagram for intents alignment of states in two different trajectories (in blue and red). **Left:** Stage I: projection into intent space (denoted by $\psi$ in text) (first 3 principal components are shown). **Middle:** Stage II: computation of intrinsic rewards for offline RL, where $s_i^a$ and $s_j^e$ are the expert's and the agent's states, $P$ is the optimal coupling matrix; corresponding intrinsic reward $r(s_i^a)$ is a scaling transform of the product $\sum_j P_{ij} C_{ij}$ with some cost function $C$ defined on the intents pairs. **Right:** squared norm *vs.* steps count in the same trajectory for the states and the intents differences; It demonstrates that distance between intents is proportional to the total path length (steps count) between the states (AntMaze example is shown).

(Touati & Ollivier, 2021) with their extensions to the generalized value functions (Hansen et al., 2019; Touati et al., 2022; Ghosh et al., 2023; Bhateja et al., 2023). In the same vein, we learn successor features but enforce additional constraint on temporal structure. Finding optimal temporal goal-conditioned value function $V^*$ (Wang et al., 2023b) which depicts the minimal number of steps, required to arrive at certain goal $g$ from state $s$ to $s_+$ can be viewed as goal conditioned RL maximizing $r(s, g) = -\mathbb{1}(s \neq g)$. ICVF (Ghosh et al., 2023) exchanges $g$ with intent $z$, generalizing previous definition of optimal goal-conditioned value function as

$$V^*(s, s_+, z) = \mathbb{E}_{s_{t+1} \sim P_z(\cdot | s_t)} \left[ \sum_{t \geq 0} -\mathbb{1}(s_t \neq s_+) \Big| s_0 = s \right], \tag{4}$$

where intent $z$ completely specifies the state-occupancy dynamics through the transition function $P_z$. In practice, $V$ is parametrized by three neural networks with $\phi, \psi : \mathcal{S} \to \mathbb{R}^d$ and $T : \mathbb{R}^d \to \mathbb{R}^{d \times d}$ being a transition matrix, incorporating all possible transition dynamics of each $z$:

$$V(s, s_+, z) = \phi(s)^T T(z) \psi(s_+). \tag{5}$$

During training, the following temporal distance loss is minimized:

$$\mathcal{L}_\tau^2(-\mathbb{1}(s \neq s_+) + \gamma \bar{V}(s', s_+, z) - V(s, s_+, z)), \tag{6}$$

with $\mathcal{L}_\tau^2(x) = |\tau - \mathbb{1}(A < 0)|x^2$, $A = r_z(s) + \gamma \bar{V}(s', s_+, z) - \bar{V}(s, s_+, z)$ being an expectile loss (Kostrikov et al., 2021), $A$ is an advantage of acting according to $z$, and $\bar{V}$ is a 'delayed' copy (target network) of $V$. Such a representation estimates an average path length between the states $s$ and $s_+$, conditioning on guidance of all possible choices of $z$. The pretraining procedure enables a downstream extraction of learned representations $\psi, \phi$ useful for the other tasks. In our experiments, we set intent of a state $s$ as $z(s) = \psi(s)$ and $r_z(s) = -\mathbb{1}(s \neq s_z)$ with a randomly sampled future state $s_z$ from the same trajectory. The $\psi$ mapping itself (without $\phi$ and $T$) serves as a good estimate for the distances between the states of the environment. In the Experiments section below, we will empirically support this, showing that a squared distance $d(s_{t+k}, s_t) = \|\psi(s_{t+k}) - \psi(s_t)\|^2$ is roughly a linear function of the steps count between the states $k$, which maps temporally similar states in the initial state space to the spatially closest in the intent space. Thus, applying optimal transport tools in this space makes it robust to noise or inaccuracies in the initial state space of the environment. Extracting the intents from the pretrained general-purpose value function enables capturing *actual behavior* from the trajectory.

We also provide a theoretical confirmation of the convergence of extracted intents to the temporal distance function between states (ref. A.1), showing explicitly that if the intentions of states $s$ and $s_+$ converges to each other, then the function $V(s, s_+)$ approaches zero in the limit. We proved theoretically (Proposition 1) and experimentally (Figure 3) that the chosen metric corresponds to the stated hypothesis (the distance function between states estimates the minimum number of steps between them). Such results are not presented in previous works. This is a new property and a new application of the "intents", which we investigated in this paper.

## 4 METHOD

Opposite to Luo *et al.* (Luo et al., 2023), we change perspective of aligning occupancy measures from the state space into the alignment of intents in the metric-aware latent space by performing isometric transformation by terms of temporally preserving function (Section 3.3). First, we briefly outline how the optimal transportation is computed in the state space.

For two trajectories $\{s_i^a\}_{i=1}^{T_a}$ and $\{s_j^e\}_{j=1}^{T_e}$, the optimal transition matrix is obtained by solving the entropy-regularized (Sinkhorn) (Cuturi, 2013) OT problem:

$$P^* = \underset{P \in \Pi[T_a, T_e]}{\arg \min} \left\{ \sum_{ij} P_{ij} C_{ij} + \varepsilon \sum_{ij} P_{ij} \log P_{ij} \right\}, \tag{7}$$

where $C_{ij} = c(s_i^a, s_{\min(i+k, T_a)}^a, s_j^e, s_{\min(j+k, T_e)}^e)$ is some cost function with the parameter $k > 0$ and $P$ has the following marginal distributions $\forall i, j$: $\sum_{j=1}^{T_e} P_{ij} = 1/T_a$, $\sum_{i=1}^{T_a} P_{ij} = 1/T_e$, which we take to be uniform across both trajectories. By finding the optimal matrix $P^*$, we can determine the reward for each state of the trajectory $\{s_i^a\}_{i=1}^{T_a}$ as

$$r(s_i^a) = \alpha \exp \left( -\tau T_a \sum_{j=j_1}^{T_e} P_{ij}^* C_{ij} \right), \tag{8}$$

where

$$j_1 = \arg \min_j C_{1j}, \tag{9}$$

and the exponent function with a scalar hyperparameters $\alpha$ and $\tau$ serves as an additional scaling factor to diminish the impact of the states with a large total cost. The negative sign ensures that, in the process of maximizing the sum of rewards, we are effectively minimizing the optimal transport (OT) distance.

We proceed to estimate the transition matrix $P^*$ between each trajectory of the agent and the expert's demonstrations (which may consist of one or more trajectories) as described in Section 3.2. For enhanced alignment, we selectively consider the tail of the expert's trajectory $j_1, \ldots, T_e$, focusing on the nearest first states to the agent's starting position according to the cost matrix $C$. The maximum reward is selected across different trajectories when the expert provides multiple trajectories. The aggregated rewards for each trajectory of the agent are then incorporated into the offline dataset for subsequent RL training, with the policy trained by the IQL offline RL algorithm (Kostrikov et al., 2021). We use ICVF representations only in rewards definition and employ IQL without any modifications. A comprehensive recipe for the rewards computation is shown in Algorithm 1.

*Unlike the OTR method (Luo et al., 2023) which computes distances directly between the states, our approach measures differences between intentions $\psi(s)$.* Specifically, we propose to incorporate the following cost matrix:

$$C_{ij} = \|\psi(s_i^a) - \psi(s_j^e)\|^2 + \|\psi(s_{\min(i+k, T_a)}^a) - \psi(s_{\min(j+k, T_e)}^e)\|^2. \tag{10}$$

The second term in the cost function is necessary for an ordered comparison of the trajectories. In imitation learning, we always compare distributions of pairs of the expert and the agent states. This is so because we want to be not only in the same state, but we also want to act in a manner similar to the expert. Hence, during training, we enforce the distribution of the agent's intent pairs to converge to the empirical measure of the intent pairs of the expert. The motivation behind the extraction of the behaviors comes from the observation that the time-step differences in the state space provide

---

**Algorithm 1** AILOT Training

---

**Input**: $\mathcal{D}_e = \{\mathcal{T}_i^e\}_{i=1}^K$ – expert trajectories (states only);
$\mathcal{D}_a = \{\mathcal{T}_i^a\}_{i=1}^L$ – reward-free offline RL dataset
**Parameters**: $\alpha$, $\tau$ – scaling coefficients, $k$ – intents pair shift, $\varepsilon$ – OT entropy coefficient

1: Train $\psi$ intents mapping by ICVF procedure (7).
2: Let `rewards = List();`
3: **for** $\mathcal{T}^a = \{s_i^a\}$ **in** $\mathcal{D}_a$ **do**
4:     Let `R = Array();`
5:     **for** $\mathcal{T}^e = \{s_j^e\}$ **in** $\mathcal{D}_e$ **do**
6:         **for** $1 \leq i \leq T_a$ and $1 \leq j \leq T_e$ **do**
7:             Set $i\_next = \min(i+k, T_a)$ and $j\_next = \min(j+k, T_e)$;
8:             Compute cost $C_{ij} = \|\psi(s_i^a) - \psi(s_j^e)\|^2 + \|\psi(s_{i\_next}^a) - \psi(s_{j\_next}^e)\|^2$;
9:         **end for**
10:        Let $j_1 = \arg\min_j C_{1j}$;
11:        Solve $\mathrm{OT}_\varepsilon \left( \{s_i^a\}_{i=1}^{T_a}, \{s_j^e\}_{j=j_1}^{T_e} \right)$ and obtain transport matrix $P^*$;
12:        For $1 \leq i \leq T_a$ compute rewards $r(s_i^a) = \alpha \exp\left( -\tau T_a \sum_{j=j_1}^{T_e} P_{ij}^* C_{ij} \right)$;
13:        Append $\{r(s_i^a)\}_{i=1}^{T_a}$ to `R`.
14:    **end for**
15:    For each $i$ append $r_i = \min_t \mathrm{R}_{ti}$ to `rewards` (the min by expert trajectories).
16: **end for**
17: **return** `rewards`

---

erroneous values, not capturing temporal dependencies between the dataset states, which we show in Figure 3, and thus, the geometry of the environment is not represented properly, making it hard for the optimal transport to scale in high-dimensional problems.

## 5 EXPERIMENTS

In this Section, we demonstrate the performance of AILOT on several benchmark tasks, report an ablation study on varying the number of provided expert demonstrations, and provide the implementation details. First, we empirically show the ability of proposed method to efficiently utilize expert demonstrations in *sparse-reward tasks* (such as Antmaze and Adroit environments) and improve learning ability of Offline RL algorithms by providing geometrically aware dense intrinsic reward signal to agent's transitions. Second, we evaluate AILOT in the *offline Imitation Learning* (IL) setting, outperforming the state-of-the art offline IL algorithms on MuJoco locomotion tasks. Also, we include additional experiments with custom behaviors, *e.g.*, the expert hopper performing a backflip, when the agent dataset consists of completely random behaviors.

### 5.1 IMPLEMENTATION DETAILS

Dense reward relabelling by AILOT is completely decoupled from the offline policy training. In all our experiments, we endow AILOT with Implicit Q-Learning (IQL) (Kostrikov et al., 2021), which is a simple and a robust offline RL algorithm. In our additional experiments, we also test AILOT+IQL against Diffusion-QL (Wang et al., 2022), which is a recent state-of-the-art approach for offline RL.

We implement AILOT in JAX (Bradbury et al., 2018) and use the official IQL implementation[1]. Our code is written using Equinox library (Kidger & Garcia, 2021). To compute optimal transport, we use implementation of Sinkhorn algorithm (Cuturi, 2013) from OTT-JAX library (Cuturi et al., 2022). Additional details on chosen hyperparameters and settings are provided in the Appendix A.2. Each dataset includes $10^6$ samples. We do pre-training of the ICVF procedure for $250 * 10^3$ steps. The IQL parameters follow original paper recommendations (particularly $10^6$ train steps with batch size 256).

---

[1]https://github.com/ikostrikov/implicit_q_learning

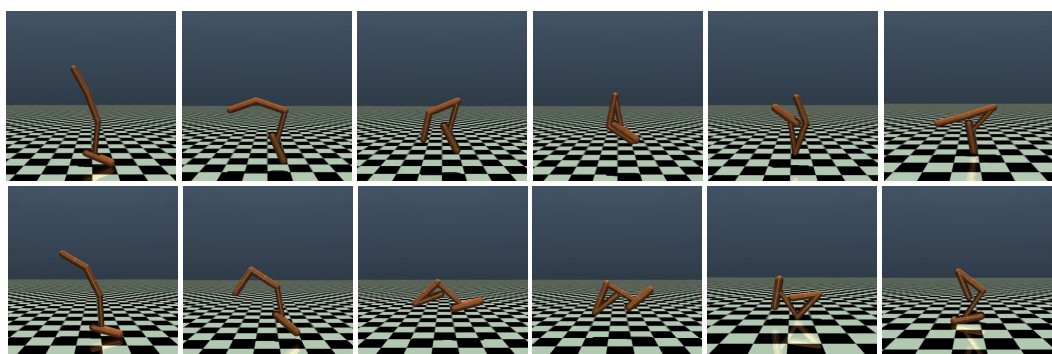

Figure 2: **Top**: Sample trajectory from the agent dataset, showcasing backflip task. **Bottom**: Hopper agent successfully performs imitation of backflip from the observations via AILOT. Refer to the Supplementary material for the animations.

**Runtime**    All of the experiments were conducted on a single RTX 3090. The runtime overhead of the application of optimal transport is not greater than extra $\sim 10$ minutes (compared to the offline RL algorithm training time). Thus, the total runtime of AILOT+IQL equals $\sim 25$ minutes. Finding optimal matrix $P^*$ and correspondent internal rewards takes several hundred iterations of Sinkhorn solver, which is relatively fast. Moreover, because the optimal transport is computed in a fixed latent space of intents, the overhead stays constant across broad range of tasks regardless of the state dimensionality. For high-dimensional tasks (*e.g.*, learning from pixels), AILOT is more suitable than OTR due to performing OT in the latent space, accelerating the overall relabelling.

## 5.2    RESULTS & EXPERIMENTS

**Baselines**    AILOT is compared against the following state-of-the-art algorithms: **IQL** (Kostrikov et al., 2021), **OTR** (Luo et al., 2023), **CLUE** (Liu et al., 2023), **IQ-Learn** (Garg et al., 2021), **Diffusion-QL** (Wang et al., 2022), **SQIL** (Reddy et al., 2019), **ORIL** (Zolna et al., 2020), and **SMODICE** (Ma et al., 2022a). The details are given in Appendix B.

We present the results for the following offline RL settings: (1) the imitation setting, where the goal is to mimic the behavior as closely as possible given several expert demonstrations in the form of trajectories; (2) the sparsified reward tasks, where the reward of one is given only when the agent reaches the goal state. We show that the rewards obtained through AILOT relabelling in both settings are descriptive enough to recover the demonstrated policy. We evaluate proposed method on common D4RL (Fu et al., 2020) benchmark and consider following domains: Gym, Adroid and Antmaze. Additional details on the environments are included in the Appendix F.

**Offline Imitation Learning.**    We compare AILOT performance on IL tasks in the offline setting. D4RL MuJoco locomotion datasets are used as the main source of offline data with the original reward signal and the action labels discarded, and the expert trajectories chosen for each task as the best episodes achieving maximal return. Results for D4RL locomotion tasks are presented in Table 1. AILOT achieves the best performance in 7 out of 9 benchmarks, when compared to OTR and CLUE. Note that, unlike the CLUE method, our algorithm does not rely on the action labels. Empirically, we confirm that the optimal geometrically aware map between the expert and the agent trajectories in the intents latent space gives reasonable guidance for the agent to learn properly. Since representation $\psi(s)$ was pretrained by general objective Eq.5 for all possible $z$, we can extract it to obtain semantic behavior of policy, which produced high return expert trajectories. We chose Euclidean distance as a measure of similarity in the intent latent space as in Eq.10 and perform exponential scaling of rewards according to Eq.8 in order to maintain them in the appropriate range.

Also, we include additional imitation results when a custom demonstrations are provided, as in Figure 2 and Figure 5, where the Hopper expert makes a backward flip and the HalfCheetah stands upwards respectively. For those tasks, agent's dataset initially consists of only the random behaviors and agent learns to recover the desired behavior through the AILOT dense relabelling.

| Dataset | IQ-Learn | SQIL | ORIL | SMODICE | **AILOT** |
|---|---|---|---|---|---|
| halfcheetah-medium-v2 | $21.7 \pm 1.5$ | $24.3 \pm 2.7$ | $56.8 \pm 1.2$ | $42.4 \pm 0.6$ | $47.7 \pm 0.2$ |
| halfcheetah-medium-replay-v2 | $6.7 \pm 1.8$ | $43.8 \pm 1.0$ | $46.2 \pm 1.1$ | $38.3 \pm 2.0$ | $42.4 \pm 0.2$ |
| halfcheetah-medium-expert-v2 | $2.0 \pm 0.4$ | $6.7 \pm 1.2$ | $48.7 \pm 2.4$ | $81.0 \pm 2.3$ | $92.4 \pm 1.5$ |
| hopper-medium-v2 | $29.6 \pm 5.2$ | $66.9 \pm 5.1$ | $96.3 \pm 0.9$ | $54.8 \pm 1.2$ | $82.2 \pm 5.6$ |
| hopper-medium-replay-v2 | $23.0 \pm 9.4$ | $98.6 \pm 0.7$ | $56.7 \pm 12.9$ | $30.4 \pm 1.2$ | $98.7 \pm 0.4$ |
| hopper-medium-expert-v2 | $9.1 \pm 2.2$ | $13.6 \pm 9.6$ | $25.1 \pm 12.8$ | $82.4 \pm 7.7$ | $103.4 \pm 5.3$ |
| walker2d-medium-v2 | $5.7 \pm 4.0$ | $51.9 \pm 11.7$ | $20.4 \pm 13.6$ | $67.8 \pm 6.0$ | $78.3 \pm 0.8$ |
| walker2d-medium-replay-v2 | $17.0 \pm 7.6$ | $42.3 \pm 5.8$ | $71.8 \pm 9.6$ | $49.7 \pm 4.6$ | $77.5 \pm 3.1$ |
| walker2d-medium-expert-v2 | $7.7 \pm 2.4$ | $18.8 \pm 13.1$ | $11.6 \pm 14.7$ | $94.8 \pm 11.1$ | $110.2 \pm 1.2$ |
| D4RL Locomotion total | 122.5 | 366.9 | 433.6 | 541.6 | 732.8 |

| Dataset | IQL | OTR | CLUE | **AILOT** |
|---|---|---|---|---|
| halfcheetah-medium-v2 | $47.4 \pm 0.2$ | $43.3 \pm 0.2$ | $45.6 \pm 0.3$ | $47.7 \pm 0.35$ |
| halfcheetah-medium-replay-v2 | $44.2 \pm 1.2$ | $41.3 \pm 0.6$ | $43.5 \pm 0.5$ | $42.4 \pm 0.8$ |
| halfcheetah-medium-expert-v2 | $86.7 \pm 5.3$ | $89.6 \pm 3.0$ | $91.4 \pm 2.1$ | $92.4 \pm 1.54$ |
| hopper-medium-v2 | $66.2 \pm 5.7$ | $78.7 \pm 5.5$ | $78.3 \pm 5.4$ | $82.2 \pm 5.6$ |
| hopper-medium-replay-v2 | $94.7 \pm 8.6$ | $84.8 \pm 2.6$ | $94.3 \pm 6.0$ | $98.7 \pm 0.4$ |
| hopper-medium-expert-v2 | $91.5 \pm 14.3$ | $93.2 \pm 20.6$ | $96.5 \pm 14.7$ | $103.4 \pm 5.3$ |
| walker2d-medium-v2 | $78.3 \pm 8.7$ | $79.4 \pm 1.4$ | $80.7 \pm 1.5$ | $78.3 \pm 0.8$ |
| walker2d-medium-replay-v2 | $73.8 \pm 7.1$ | $66.0 \pm 6.7$ | $76.3 \pm 2.8$ | $77.5 \pm 3.1$ |
| walker2d-medium-expert-v2 | $109.6 \pm 1.0$ | $109.3 \pm 0.8$ | $109.3 \pm 2.1$ | $110.2 \pm 1.2$ |
| D4RL Locomotion total | 692.4 | 685.6 | 714.5 | 732.8 |

Table 1: Normalized scores (mean ± standard deviation) of AILOT on MuJoco locomotion tasks, compared to baselines. The upper sub-table includes methods for imitation learning without any kind of optimal transport. The lower sub-table shows results for the conceptually close approaches – oracle IQL, OTR, and CLUE. For these methods the results are given for $K = 1$ number of expert trajectories. The highest scores are highlighted in green.

| Dataset | IQL | OTR | CLUE | **AILOT** |
|---|---|---|---|---|
| umaze-v2 | 88.7 | $81.6 \pm 7.3$ | $92.1 \pm 3.9$ | $93.5 \pm 4.8$ |
| umaze-diverse-v2 | 67.5 | $70.4 \pm 8.9$ | $68.0 \pm 11.2$ | $63.4 \pm 7.6$ |
| medium-play-v2 | 72.9 | $73.9 \pm 6.0$ | $75.3 \pm 6.3$ | $71.3 \pm 5.2$ |
| medium-diverse-v2 | 72.1 | $72.5 \pm 6.9$ | $74.6 \pm 7.5$ | $75.5 \pm 7.4$ |
| large-play-v2 | 43.2 | $49.7 \pm 6.9$ | $55.8 \pm 7.7$ | $57.6 \pm 6.6$ |
| large-diverse-v2 | 46.9 | $48.1 \pm 7.9$ | $49.9 \pm 6.9$ | $66.6 \pm 3.1$ |
| AntMaze-v2 total | 391.3 | 396.2 | 415.7 | 427.9 |

Table 2: Normalized scores (mean ± standard deviation) of AILOT on sparse-reward environments. We compare different dense relabelling methods (OTR, CLUE) and show that AILOT outperforms those approaches and accelerates learning of offline IQL method.

**Sparse-Reward Offline RL Tasks.** Next, we evaluate AILOT on several sparse D4RL benchmarks (AntMaze-v2 and Adroit-v2). To obtain expert trajectories, we consider only those episodes that accomplish the goal task, dismissing all the others.

Table 3 compares the performance of AILOT+IQL to OTR+IQL, CLUE+IQL, when only a single demonstration trajectory is available, and to the original IQL. We employed IQL, which acts as a baseline, without any modifications. Given that OTR + IQL always performs better than IQL, we decided to compare directly with OTR, which showcases the importance of performing optimal

| Dataset | IQL | OTR | CLUE | **AILOT** |
|---------|-----|-----|------|-----------|
| door-cloned-v0 | 1.6 | $0.01_{\pm 0.01}$ | $0.02_{\pm 0.01}$ | $0.05_{\pm 0.02}$ |
| door-human-v0 | 4.3 | $5.9_{\pm 2.7}$ | $7.7_{\pm 3.9}$ | $7.9_{\pm 3.2}$ |
| hammer-cloned-v0 | 2.1 | $0.9_{\pm 0.3}$ | $1.4_{\pm 1.0}$ | $1.6_{\pm 0.1}$ |
| hammer-human-v0 | 1.4 | $1.8_{\pm 1.4}$ | $1.9_{\pm 1.2}$ | $1.8_{\pm 1.3}$ |
| pen-cloned-v0 | 37.3 | $46.9_{\pm 20.9}$ | $59.4_{\pm 21.1}$ | $61.4_{\pm 19.5}$ |
| pen-human-v0 | 71.5 | $66.8_{\pm 21.2}$ | $82.9_{\pm 20.2}$ | $89.4_{\pm 0.1}$ |
| relocate-cloned | -0.2 | $-0.24_{\pm 0.03}$ | $-0.23_{\pm 0.02}$ | $-0.20_{\pm 0.03}$ |
| relocate-human | 0.1 | $0.1_{\pm 0.1}$ | $0.2_{\pm 0.3}$ | $0.28_{\pm 0.1}$ |
| Adroit-v0 total | 118.1 | 122.2 | 153.3 | 162.2 |

Table 3: Normalized scores (mean $\pm$ standard deviation) of AILOT on Adroit and AntMaze tasks, compared to baselines. OTR, CLUE, and AILOT use IQL as offline RL baseline algorithm with only a single expert episode. The highest scores are highlighted.

transport alignment under distance preserving mapping. We observe that AILOT outperforms the current state-of-the art results, with the hardest antmaze-large-diverse task showing the most remarkable margin. This proves the ability of AILOT to employ the expressive representations from general pretrained value function (5) through the optimal transport for functional learning in the sparse tasks as well. In door-cloned-v0 and hammer-cloned-v0 tasks IQL performs best with sparse rewards. The inferior results in those two tasks stem from the fact that the cloned version was obtained from collecting the data from trained imitation learning policy on a mix of human and the expert data. Human data contains many ambiguous trajectories, from which it is hard to extract valuable intents (this is the case for both the door and the hammer-cloned tasks).

## 5.3 ABLATION STUDY

**Varying the number of expert trajectories.** We investigate whether performance of learned behavior tends to improve with increased number of provided expert trajectories. Table 7 in the Appendix shows overall performance for varying number of expert trajectories for $K = 1$ to $K = 5$ across OTR and AILOT. However, we observe that performance across both algorithms improves slightly. We make comparison with OTR here because it's the most similar to ours (it also uses OT for reward labelling and IQL for RL problem solution). Still, AILOT achieves better normalized scores than OTR, thus proving that alignment of intents in geometry-aware space improves intrinsic rewards labelling in comparison with similar rewards but with pairwise distances between original states as done in OTR.

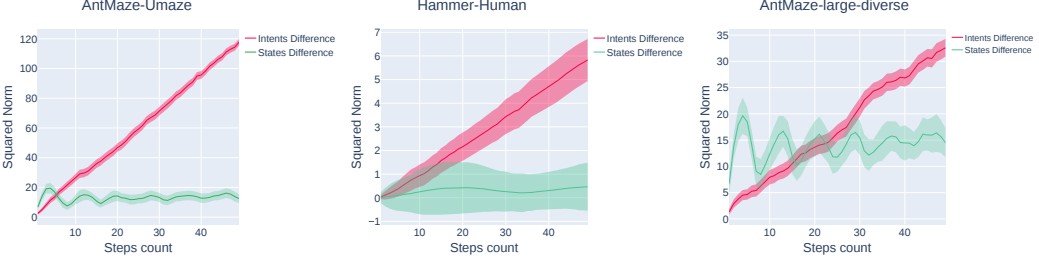

Figure 3: The squared norm of states and intents differences, depending on the total steps count between the states in the same trajectory. The intents differences $||\psi(s_{t+k}) - \psi(s_t)||^2$ has a near-linear dependence on the steps count. The squared norm in the state space is not a monotone function, which is less efficient for training an imitating agent, since it completely ignores global geometric dependencies between states in the dataset.

**Intents distance dependence on the steps count.** By learning temporally preserving value function (i.e., similar states get mapped to temporally closed points in the intents space), the underlying space

becomes more structured. Because the optimal transport takes the geometry of space into account, the task of finding the alignment becomes simpler. In the Method section, we have already mentioned that $\mathbb{E}_t \|\psi(s_{t+k}) - \psi(s_t)\|^2$ is near-linearly dependent on the steps count between the states ($k$). This important property plays a constructive role in the cost matrix in the OT problem (10), and ultimately, gives a good estimate of the distance between the trajectories. Empirical evidence of the near-linear dependence is presented in Figure 3. On this figures we show that the pairwise distances between the original states $\mathbb{E}_t \|s_{t+k} - s_t\|^2$, which is what used in previous OT methods, have no such feature and their direct use as a cost function is less preferable, since they discarding global temporal geometry. Additionally, we compare AILOT to OTR with the same $\alpha$ and $\beta$ hyperparameters in the Appendix, showcasing that performance gains come from performing OT in temporally grounded latent space.

## 6 DISCUSSION

In our work, we introduced AILOT – a new non state-occupancy estimation method for extracting expert's behavior in terms of its *intentions* and guiding the learning agent to them through an intrinsic reward. We empirically show that it surpasses the state-of-the art results both in the sparse-reward RL tasks and in the offline imitation learning setting. AILOT can mimic the expert behaviour without knowing its action labels, and without the ground truth rewards. Moreover, we show that the intrinsic rewards, distilled by AILOT, could be used to efficiently boost the performance of other offline RL algorithms, thanks to the proper alignment to the expert intentions via the optimal transport.

**Limitations** of AILOT include assumption on sufficient access to a large number of unlabeled trajectories of some *acceptable* quality. In our work, we have focused on the expert behaviours, typically considered in the offline RL publications: popular synthetic environments with some comprehensible expert movements and, consequently, some sufficiently 'intuitive' intentions, which can be easily extracted from the provided datasets.

Another limitation follows from the multi-modality of intents, because the expert can have several goals or performs a vague action. The imitation efficiency could, of course, drop, because the expert intents may no longer be transparent to the agent. While such a trait would be on par with the way humans learn a certain skill by observing an adept, weighing the hierarchy of multiple possible intents in AILOT could prove useful to further regularize the learning dynamics in such uncertain scenarios and is an interesting direction for further research. Other direction of future work is to venture into the *cross-domain imitation*. Based on the results observed here, it should be possible to generalize AILOT to handle the transition shift between the expert and the agent in the presence of larger mismatches pertinent to the different domains.

In conclusion, the development of AILOT sets a robust benchmark for future generalizations, enhancing ongoing research in crafting generalist knowledge distillation agents from the offline data.

**Reproducibility Statement** We included source code in the supplementary material. All of the hyperparameters used, along with additional dataset preprocessing step, are discussed in Appendix A.2. In our experiments, we used open-source D4RL datasets (Fu et al., 2020). In order to ensure reproducibility and account for randomness, we repeated our experiments over 10 random seeds.

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

# A APPENDIX

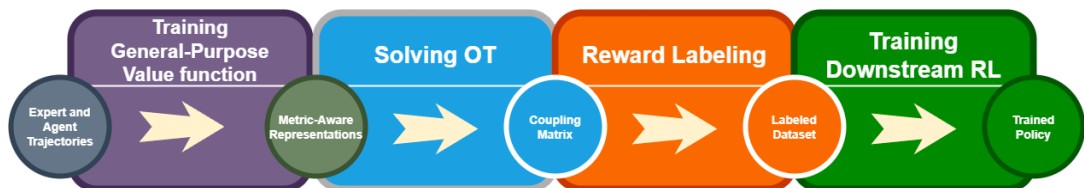

Figure 4: Principal diagram of AILOT approach for imitation learning.

## A.1 VALUE FUNCTION CONVERGENCE

**Proposition 1** *Consider all possible pairs of current and goal states $(s, s_+)$ with $z = \psi(s_+)$. Assume that functions $\phi(s)$, $\psi(s_+)$, $T(\psi(s_+))$ (decomposition of $V(s, s_+, z)$ from eq. 5) and expression*

$$\left\| \phi^T(s) \frac{\partial T(\psi_0)}{\partial \psi} \psi(s_+) \right\| \tag{11}$$

*are upper-bounded by some constant value. Here $\psi_0$ is some intermediate value between $\psi(s)$ and $\psi(s_+)$. Then from convergence of intent embedding $\psi(s) \to \psi(s_+)$ (w.r.t Euclidean norm) it follows*

$$V(s, s_+, z) \to 0.$$

*Proof*: Define $V(s, s_+) = V(s, s_+, \psi(s_+))$ and $\delta = \|\psi(s) - \psi(s_+)\|$. The goal is to find an upper bound $V(s, s_+)$. Since $V(s, s) = 0$, $-V(s, s_+)$ can be rewritten as

$$-V(s, s_+) = -V(s, s_+) + V(s, s). \tag{12}$$

Take by definition $V(s, s_+) = \phi^T(s) T(\psi(s_+)) \psi(s_+)$ and obtain

$$-V(s, s_+) = \phi^T(s) T(\psi(s)) \psi(s) - \phi^T(s) T(\psi(s_+)) \psi(s_+) \tag{13}$$

$$= \phi^T(s) T(\psi(s)) [\psi(s) - \psi(s_+)] + \phi^T(s) [T(\psi(s)) - T(\psi(s_+))] \psi(s_+) \tag{14}$$

$$\leq \|\phi^T(s) T(\psi(s))\| \delta + \left\| \phi^T(s) \frac{\partial T(\psi_0)}{\partial \psi} \psi(s_+) \right\| \delta. \tag{15}$$

Since the norms in the expression above are upper-bounded then from $\delta \to 0$ follows $V(s, s_+) \to 0$.

## A.2 HYPERPARAMETERS

We found that $n = 200$ Sinkhorn solver iterations are enough to find optimal coupling and set $\epsilon = 0.001$ as entropy regularization parameter. All hyperparameters for IQL on D4RL benchmarks are set to those recommended in original paper to ensure reproducibility. We use default latent dimension of $d = 256$ for general value function from Eq. 5 embeddings pretraining across all benchmarks and set other hyperparameters as in original paper. In order to maintain rewards in a reasonable range, they are scaled by exponential function as written in Equation 8 with $\alpha = 5, \tau = 0.5$ for MuJoco and AntMaze tasks and $\alpha = 5, \tau = 10$ for Adroit. We found that lookahead parameter in Eq. 10 works best for $k = 2$. Results are evaluated across 10 random seeds and 10 evaluation episodes for each seed in order to be consistent with previous works.

It should be noted that IQL incorporates its own rewards rescaling function within the dataset. We apply similar technique, using the reward scaling factor of $\dfrac{1000}{\text{max\_return} - \text{min\_return}}$.

For AntMaze-v2 tasks, we subtract 1 from the rewards outputted by AILOT. All other parameters for IQL for other tasks are kept intact. Full list of crucial parameters are presented in Tables 4 and 5. For AntMaze tasks, the parameters are the same as for Mujoco, except for the expectile in IQL, which we set to 0.9. Pretraining of general value function is executed for 250k steps, which we found to be enough to provide reasonable distance estimation, which coincides with original ICVF implementation.

| Algorithm | Hyperparameter | Value |
|---|---|---|
| IQL | Temperature | 6 |
| | Expectile | 0.7 |
| | Hidden layers | (256, 256) |
| | Optimizer | Adam |
| | Critic learning rate | $3e^{-4}$ |
| | Value learning rate | $3e^{-4}$ |
| | Policy learning rate | $3e^{-4}$ |
| AILOT | $\tau$ | 0.5 |
| | $\alpha$ | 5 |
| | Sinkhorn $\epsilon$ | 0.001 |
| | k | 2 |

Table 4: Hyperparameters for MuJoco tasks.

| Algorithm | Hyperparameter | Value |
|---|---|---|
| IQL | Temperature | 5 |
| | Expectile | 0.7 |
| AILOT | $\tau$ | 10 |
| | $\alpha$ | 5 |
| | Sinkhorn $\epsilon$ | 0.001 |
| | k | 2 |

Table 5: Hyperparameters for Adroit tasks. IQL parameters are the same as for the locomotion tasks.

## B    DETAILS ABOUT THE BASELINE MODELS

Performance of AILOT + IQL is compared to the following algorithms:

- **IQL** (Kostrikov et al. (2021)) is state-of-the-art offline RL algorithm, which avoids querying out of the distribution actions by viewing value function as a random variable, where upper bound of uncertainty is controlled through expectile of distribution. In our experiments, evaluation is made using ground-truth reward from D4RL tasks.

- **OTR** (Luo et al. (2023)) is a reward function algorithm, where reward signal is based on optimal transport distance between states of expert demonstration and reward unlabeled dataset.

- **CLUE** (Liu et al. (2023)) learns VAE calibrated latent space of both expert and agent state-action transitions, where intrinsic rewards can be defined as distance between agent and averaged expert transition representations.

- **IQ-Learn** (Garg et al. (2021)) is an imitation learning algorithm, which implicitly encodes into learned inverse Q-function rewards and policy from expert data.

- **Diffusion-QL** (Wang et al. (2022)) is a state-of-the-art offline RL algorithm, which models learning policy as conditional diffusion model in order to effectively increase expressiveness and provide more flexible regularization towards behavior policy, which collected dataset.

- **SQIL** (Reddy et al. (2019)) proposes to learn soft Q-function by setting expert transitions to one and for non-expert transitions to zero.

- **ORIL** (Zolna et al. (2020)) utilizes discriminator network which distinguishes between optimal and suboptimal data in mixed dataset to provide reward relabelling through learned discriminator.

- **SMODICE** (Ma et al. (2022a)) offline state occupancy matching algorithm, which solves the problem of IL from observations through state divergence minimization by utilizing dual formulation of value function.

## C   EXTRA ABLATION STUDY: VARYING NUMBER OF EXPERT TRAJECTORIES

In this section, we report an ablation study on varying the number of provided expert trajectories, $K$. Table 7 compares AILOT and OTR for $K = 1, 5$. We observe that the extraction of intent representations is crucial for a good performance. We also observed a negligible performance improvement when the number of provided expert trajectories exceeds certain threshold ($K \geq 10$).

| Dataset/Method | OTR | AILOT | OTR | AILOT |
|---|---|---|---|---|
| **# of expert trajectories** | $K = 1$ | $K = 1$ | $K = 5$ | $K = 5$ |
| halfcheetah-medium-v2 | $43.3_{\pm 0.2}$ | $47.7_{\pm 0.2}$ | $45.2_{\pm 0.2}$ | $46.6_{\pm 0.2}$ |
| halfcheetah-medium-replay-v2 | $41.3_{\pm 0.6}$ | $42.4_{\pm 0.2}$ | $41.9_{\pm 0.3}$ | $41.2_{\pm 0.5}$ |
| halfcheetah-medium-expert-v2 | $89.6_{\pm 3.0}$ | $92.4_{\pm 1.5}$ | $89.9_{\pm 1.9}$ | $92.4_{\pm 1.0}$ |
| hopper-medium-v2 | $78.7_{\pm 5.5}$ | $82.2_{\pm 5.6}$ | $79.5_{\pm 5.3}$ | $82.5_{\pm 3.7}$ |
| hopper-medium-replay-v2 | $84.8_{\pm 2.6}$ | $98.7_{\pm 0.4}$ | $85.4_{\pm 1.7}$ | $97.4_{\pm 0.1}$ |
| hopper-medium-expert-v2 | $93.2_{\pm 20.6}$ | $103.4_{\pm 5.3}$ | $90.4_{\pm 21.5}$ | $107.3_{\pm 5.6}$ |
| walker2d-medium-v2 | $79.4_{\pm 1.4}$ | $78.3_{\pm 0.8}$ | $79.8_{\pm 1.4}$ | $80.9_{\pm 1.4}$ |
| walker2d-medium-replay-v2 | $66.0_{\pm 1.4}$ | $77.5_{\pm 3.1}$ | $71.0_{\pm 5.0}$ | $76.9_{\pm 1.6}$ |
| walker2d-medium-expert-v2 | $109.3_{\pm 0.8}$ | $110.2_{\pm 1.2}$ | $109.4_{\pm 0.4}$ | $110.3_{\pm 0.4}$ |
| D4RL Locomotion total | $685.6$ | $732.8$ | $690.6$ | $735.5$ |

Table 6: Normalized scores for D4RL locomotion tasks with varying number of expert trajectories. The highest scores are highlighted.

Also, we performed several experiments with modified cost function in OTR method, i.e., exactly the same as in our Eq. 10 with varying levels of $k$, which corresponds to intents shift in Algorithm 1. We observe that the lookahead parameter introduces only negligible improvements to OTR (up to $+0.5$), which is an indicator that the temporal grounding of OT is completely lacking in OTR.

| Dataset/Method | OTR ($k = 2$) | AILOT |
|---|---|---|
| halfcheetah-medium-v2 | $43.3_{\pm 0.1\ (+0.1)}$ | $47.7_{\pm 0.2}$ |
| halfcheetah-medium-replay-v2 | $41.6_{\pm 0.8\ (+0.4)}$ | $42.4_{\pm 0.2}$ |
| halfcheetah-medium-expert-v2 | $89.8_{\pm 3.0\ (+0.2)}$ | $92.4_{\pm 1.5}$ |
| hopper-medium-v2 | $79.3_{\pm 5.5\ (+0.6)}$ | $82.2_{\pm 5.6}$ |
| hopper-medium-replay-v2 | $85.1_{\pm 2.6\ (+0.3)}$ | $98.7_{\pm 0.4}$ |
| hopper-medium-expert-v2 | $93.5_{\pm 20.6\ (+0.2)}$ | $103.4_{\pm 5.3}$ |

Table 7: Normalized scores for D4RL locomotion tasks with varying number of expert trajectories. The highest scores are highlighted.

## D   ADDITIONAL COMPARISON EXPERIMENTS

We tested AILOT + O-DICE (instead of IQL). O-DICE by Mao et al. (2024) itself shows better scoring than IQL, and the corresponding replacement of the RL algorithm gives improvements in all the tasks below (Table 8). It justifies that our method is RL-algorithm-agnostic and can be used with any other offline RL method, for example with O-DICE. The comparison between O-DICE and AILOT+O-DICE is not quite correct as they solve different problems (offline RL and imitation learning, respectively). But it should be noted that our scores on hard antmaze-large-play-v2 and antmaze-large-diverse-v2 tasks outperform those of O-DICE. This is of no surprise since temporal grounding is lacking in the O-DICE method, where only distribution matching with behavior dataset is performed.

We also compare the performance of AILOT to Diffusion-QL and Behavior Cloning (10%), which are shown in Table 9. We evaluated Diffusion-QL with recommended parameters from the original paper and tested performance on sparse-reward AntMaze tasks.

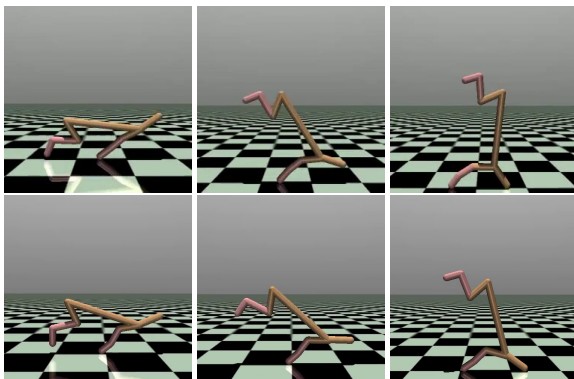

Figure 5: **Top**: HalfCheetah expert sample trajectory, performing an upwards standing which the agent should imitate; **Bottom**: HalfCheetah agent successfully performs imitation of standing upwards from the observations via AILOT.

| Dataset | O-DICE | AILOT + IQL | AILOT + O-DICE |
|---|---|---|---|
| halfcheetah-medium-v2 | $47.4_{\pm 0.2}$ | $47.7_{\pm 0.2}$ | $49.5_{\pm 0.4}$ |
| halfcheetah-medium-replay-v2 | $44.0_{\pm 0.3}$ | $42.4_{\pm 0.2}$ | $46.2_{\pm 0.6}$ |
| halfcheetah-medium-expert-v2 | $93.2_{\pm 0.6}$ | $92.4_{\pm 1.5}$ | $93.6_{\pm 1.8}$ |
| hopper-medium-v2 | $86.1_{\pm 4.0}$ | $82.2_{\pm 5.6}$ | $85.5_{\pm 3.7}$ |
| hopper-medium-replay-v2 | $99.9_{\pm 2.7}$ | $98.7_{\pm 0.4}$ | $99.1_{\pm 0.2}$ |
| hopper-medium-expert-v2 | $110.8_{\pm 0.6}$ | $103.4_{\pm 5.3}$ | $106.9_{\pm 2.1}$ |
| antmaze-large-play-v2 | $55.9_{\pm 3.9}$ | $57.6_{\pm 6.6}$ | $58.2_{\pm 4.3}$ |
| antmaze-large-diverse-v2 | $54.0_{\pm 4.8}$ | $66.6_{\pm 3.1}$ | $68.3_{\pm 3.1}$ |

Table 8: Peformance of AILOT with different offline RL methods (IQL and O-DICE) and comparison with O-DICE.

| Dataset | DiffusionQL | **AILOT** |
|---|---|---|
| antmaze-large-diverse | $56.6_{\pm 7.6}$ | $66.6_{\pm 3.1}$ |
| antmaze-large-play | $46.4_{\pm 8.3}$ | $57.6_{\pm 6.6}$ |

| Dataset | BC-10 | **AILOT** |
|---|---|---|
| halfcheetah-medium-v2 | 42.5 | $47.7_{\pm 0.2}$ |
| halfcheetah-medium-replay-v2 | 40.6 | $42.4_{\pm 0.2}$ |
| halfcheetah-medium-expert-v2 | 92.9 | $92.4_{\pm 1.5}$ |
| hopper-medium-v2 | 56.9 | $82.2_{\pm 5.6}$ |
| hopper-medium-replay-v2 | 75.9 | $98.7_{\pm 0.4}$ |
| hopper-medium-expert-v2 | 110.9 | $103.4_{\pm 5.3}$ |
| walker2d-medium-v2 | 75.0 | $78.3_{\pm 0.8}$ |
| walker2d-medium-replay-v2 | 62.5 | $77.5_{\pm 3.1}$ |
| walker2d-medium-expert-v2 | 109.0 | $110.2_{\pm 1.2}$ |

Table 9: Peformance of AILOT in comparison to Diffusion-QL on sparse-reward AntMaze task and BC-10 results.

## E    Ablating $\alpha$ and $\tau$

In the current section we provide expriments on how hyperparameters $\alpha$ and $\tau$ are influencing overall performance. Provided results are similiar to those reported in (Luo et al., 2023).

| Dataset | AILOT($\alpha = 5, \tau = 0.5$) | AILOT($\alpha = 1, \tau = 1$) |
|---|---|---|
| halfcheetah-medium-v2 | $47.7_{\pm 0.35}$ | $45.6_{\pm 0.3}$ |
| halfcheetah-medium-replay-v2 | $42.4_{\pm 0.8}$ | $40.2_{\pm 0.5}$ |
| halfcheetah-medium-expert-v2 | $92.4_{\pm 1.5}$ | $89.4_{\pm 0.7}$ |
| walker2d-medium-v2 | $78.3_{\pm 0.8}$ | $74.3_{\pm 0.4}$ |
| walker2d-medium-replay-v2 | $77.5_{\pm 3.1}$ | $71.7_{\pm 2.1}$ |
| walker2d-medium-expert-v2 | $110.2_{\pm 1.2}$ | $96.7_{\pm 1.0}$ |
| antmaze-large-play-v2 | $57.6_{\pm 6.6}$ | $56.3_{\pm 4.6}$ |
| antmaze-large-diverse-v2 | $66.6_{\pm 3.1}$ | $63.4_{\pm 2.1}$ |

Table 10: How the hyperparameters $\alpha$ and $\tau$ influence the performance.

## F    Description of Evaluated Environments

Gym-MuJoCo locomotion are the most commonly used tasks for the evaluation of performance of RL algorithms. D4RL provides precollected datasets of different quality, corresponding to training policies evaluated after training for certain number of steps.

**Gym-Locomotion.**    Locomotion tasks consist of four environments: Hopper, Walker2d, HalfCheetah, and Ant. Provided datasets from D4RL come in "-expert", "-replay", and "-random" splits. Where "-expert" split corresponds to high-reward trajectories, whilst the others contain a mixture of roll-outs from partially trained policies or even some random transitions. All of the datasets are obtained with the SAC algorithm.

**Manipulation**    In addition to the locomotion experiments, we test the proposed approach on *Adroit* environments. Adroid involves controlling a 24-DoF Hand robot tasked with opening a door, hammering a nail, picking up and moving a ball. Two types of offline data from D4RL are considered: human demonstrations (with "-human" split), which were obtained from logged experiences of humans and data obtained from a sub-optimal policy (with "-cloned" split), obtained from the imitation policy.

