# OpenReview forum: "Align Your Intents: Offline Imitation Learning via Optimal Transport"
_ICLR.cc/2025/Conference — Submitted to ICLR 2025_

### Official Review · Reviewer_dczu · 2024-11-01

**Soundness:** 2
**Presentation:** 2
**Contribution:** 2
**Rating:** 3
**Confidence:** 4

**Summary:**

This paper introduces Aligned Imitation Learning via Optimal Transport (AILOT), a method for offline reinforcement learning that uses optimal transport to align an agent's behavior with an expert's in a "intent" space. The intent space is learned with some previously suggested method. AILOT outperforms existing methods on benchmark tasks, especially in sparse-reward environments.

**Strengths:**

- Strong empirical performance

**Weaknesses:**

- This paper is basically a souped-up version of (Luo et al., 2023), where the optimal transport between state is replaced with the optimal transport between intentions. Since the intention space is also learned with the previously suggested method, the contribution of the paper is  1) idea of using intention instead of state itself, and 2) the design of the cost matrix in Eq. (10). However, the design of the cost matrix Eq. (10) is not well analyzed in the paper, neither theoretically nor empirically. In my opinion, the idea of using intention instead of state is straightforward, and the paper for ICLR should contain more messages than it.

- SInce the usage of intention on OTR is the key contribution of the paper, I would expect the paper to analyze on what aspects do we need to improve from state space OTR methods. However, the paper relies on a single intention learning method and do not discuss on why the used intention learning method improves.

**Questions:**

- The paper argues that learning expert states alone (without actions nor rewards) is one strong contribution of the paper. Are previous works incapable of doing that (e.g., (Luo et al., 2023))?

---

> ### Author Response · Authors · 2024-11-25
>
> Thank you very much for your time your thorough review.  I'll address each point carefully below.
>
> ***This paper is basically a souped-up version of (Luo et al., 2023), where the optimal transport between state is replaced with the optimal transport between intentions:***
>
> We respectfully disagree with this assessment. Our original hypothesis was to improve the distance function between states so that it estimates the minimum number of steps between states instead of the conventional Euclidean distance. In addition to evaluating the algorithm as a whole on d4lr, we also investigated the properties of the intent distance metric itself. We proved theoretically (Proposition 1) and experimentally (Figure 3) that the chosen metric corresponds to the stated hypothesis. Such results are not presented in previous works of OTR (Luo et al., 2023) and ICVF (D. Ghosh et al., 2023). This is a new property and a new application of intents, reported for the first time in our paper.
>
> ***The design of the cost matrix Eq. (10) is not well analyzed in the paper:***
>
> The design change in matrix C (eq. 10) merely consists of setting k=2 instead of k=1. We have studied the effect of this modification empirically (please refer to Table 7). As demonstrated, setting k=2 gives a small improvement. Smaller k=1 may not capture enough temporal context, while larger values k>2 make the alignment harder.
>
> ***The paper argues that learning expert states alone (without actions nor rewards) is one strong contribution of the paper:***
>
> Respectfully, this is not quite true; in lines 70-76 we list three contributions of the paper, of which the main one is a new way of specifying the intrinsic reward function. Learning without the rewards and the expert actions is a common practice in imitation learning, and we have shown how this can be done more efficiently. As a result, we report better metric scores (Table 1) than those algorithms that use traditional real rewards extracted from the environment.

---

> > ### Author Response · Authors · 2024-11-28
> >
> > Dear reviewer, please kindly let us know if we have clarified our contributions. The discussion period is about to end and we have not had a chance to elaborate anything further during the entire discussion period. Please let us know if any other items need our clarification. Thank you.

---

> > > ### Comment · Reviewer_dczu · 2024-12-01
> > >
> > > After reading the authors' response, I am still not convinced whether the contribution of the paper is sufficient. I keep my score.

---

### Official Review · Reviewer_LCQn · 2024-11-03

**Soundness:** 3
**Presentation:** 2
**Contribution:** 2
**Rating:** 5
**Confidence:** 4

**Summary:**

This paper addresses how to effectively imitate expert behavior in **offline reinforcement learning** with **sparse rewards** and without action labels or ground truth rewards. Previous methods used optimal transport to measure similarity between agent and expert trajectories as a reward signal, but relied on raw state distances. This paper introduces **AILOT**, which uses "intent alignment" and "optimal transport" in an intent space to calculate intrinsic rewards, enabling the agent to learn expert behavior more effectively and improve performance in sparse reward tasks.

**Strengths:**

1.	**Novel approach within the scope of existing methods:** While optimal transport has been used in similar imitation learning research, AILOT applies it uniquely by focusing on “intent alignment” in a metric-aware latent space, which differentiates it from other OT-based approaches.
2.	**Demonstrated performance improvement:** AILOT outperforms several baseline models, including those using OT, in various benchmark tasks, indicating it successfully optimizes OT alignment in a manner that strengthens offline RL without explicit reward signals.
3.	**Robust integration with other RL algorithms:** The method is designed to enhance the performance of other offline RL algorithms, making it versatile for broader applications.

**Weaknesses:**

1.	**Overlap with existing research:** The approach shares similarities with prior work that applies optimal transport to offline imitation learning, such as Optimal Transport for Offline Imitation Learning (arXiv:2303.13971) and Combining Expert Demonstrations in Imitation Learning via Optimal Transport (arXiv:2307.10810). These papers also use OT to create reward signals from expert trajectories, raising concerns about the novelty of AILOT’s contribution. Although AILOT introduces intent alignment as a distinct feature, further justification of how this approach advances beyond these prior works would strengthen the contribution.
2.	**Limited comparison with recent state-of-the-art methods:** The paper does not include comparisons with more recent imitation learning algorithms, such as O-DICE (ODICE: Revealing the Mystery of Distribution Correction Estimation via Orthogonal-gradient Update, arXiv:2402.00348), which have demonstrated strong performance in offline imitation learning tasks. Including such comparisons would provide a clearer understanding of AILOT’s relative performance and contributions.
3.	**Dependency on well-defined intents:** AILOT’s performance may be compromised if the expert’s behavior is ambiguous or multi-modal, making alignment challenging. This is especially relevant when handling multi-intent expert demonstrations, an area where existing OT-based methods may also encounter limitations.
4.	**Lack of clarity in training configuration:** While the paper provides an estimated runtime on an NVIDIA RTX 3090 GPU (10-25 minutes), it lacks specific details on training configurations, such as the number of samples or epochs used. Including this information would improve reproducibility and allow readers to better assess the computational efficiency of AILOT.

**Questions:**

1.	**Clarification on Dataset Size and Training Configuration:** Could the authors provide specific details on the number of samples and epochs used during training? This information would help clarify the computational efficiency of the method, beyond the hardware and runtime specifics provided.
2.	**Comparison with Modern State-of-the-Art Methods:** Have the authors considered including comparisons with more recent state-of-the-art methods in imitation learning, such as O-DICE or other recent 2024 approaches? This would offer a more comprehensive view of AILOT’s performance relative to current advancements in the field.
3. **Comprehensive Sensitivity Analysis for Cost Function and Hyperparameters:** While the paper includes a limited ablation study with only two configurations (α=5, τ=0.5 and α=1, τ=1), a broader exploration of these hyperparameters would provide a clearer picture of AILOT’s robustness. Could the authors expand the sensitivity analysis with more variations in these parameters or offer additional insights into how these choices affect the model’s performance across different tasks?

---

> ### Author Response · Authors · 2024-11-26
>
> Thank you very much for your time and detailed consideration of our paper!
>
> ***Overlap with existing research:***
>
> Our original hypothesis was to improve the distance function between states so that it estimates the minimum number of steps between states instead of the conventional Euclidean distance. In addition to evaluating the algorithm as a whole on d4lr, we also investigated the properties of the intent distance metric itself. We proved theoretically (Proposition 1) and experimentally (Figure 3) that the chosen metric corresponds to the stated hypothesis. Such results are not presented in previous works of OTR (Luo et al., 2023) and ICVF (D. Ghosh et al., 2023). This is a new property and a new application of intents, which we investigated in this paper.
>
> ***AILOT’s performance may be compromised if the expert’s behavior is ambiguous or multi-modal, making alignment challenging:***
>
> According to algorithm 1 (step 15) we take the minimum over expert trajectories. That is, in particular, it searches for the closest expert trajectory to the agent's trajectory. If there are expert trajectories that take different behaviors to solve the multi-modal problem then AILOT will imitate all these different behaviors.
>
> ***Clarification on Dataset Size and Training Configuration:***
>
> The paper uses D4RL datasets for evaluation, which include: Gym-MuJoCo locomotion tasks, Adroit manipulation tasks and AntMaze navigation tasks. Each dataset includes approximately 10^6 transitions (we took datasets provided in D4RL).  We do pre-training of ICVF procedure (D. Ghosh et al., 2023) for around 250k steps. In order to report final numbers, we take original IQL hyperparameters parameters (particularly 10^6 train steps with batch size 256) in order to be consistent with original IQL paper
>
> ***Comparison with Modern State-of-the-Art Methods (O-DICE):***
>
> Our method is not tied to IQL and can be used with any other offline RL method, for example with O-DICE. We tested AILOT + O-DICE (instead of IQL) and results showcase that on long-horizon planning tasks the proposed approach also outperforms O-DICE. O-DICE itself shows better scoring than IQL, and the corresponding replacement of the RL algorithm gives improvements in almost all the tasks below.
>
>
> | Dataset               | O-DICE     | AILOT + IQL      | AILOT + O-DICE |
> | --------------------- | ---------- | ---------- | -------------- |
> | halfcheetah-medium-v2         | 47.4 &pm; 0.2  | 47.7 &pm; 0.2 | **49.5 &pm; 0.4**     |
> | halfcheetah-medium-replay-v2       | 44.0  &pm; 0.3 | 42.4 &pm; 0.2  | **46.2 &pm; 0.6**    |
> | halfcheetah-medium-expert-v2       | 93.2 &pm; 0.6  | 92.4 &pm; 1.5  | **93.6 &pm; 1.8**   |
> | hopper-medium-v2              | **86.1 &pm; 4.0**  | 82.2 &pm; 5.6  | 85.5 &pm; 3.7      |
> | hopper-medium-replay-v2             | **99.9 &pm; 2.7**  | 98.7 &pm; 0.4  | 99.1  &pm; 0.2     |
> | hopper-medium-expert-v2           | **110.8 &pm; 0.6** | 103.4 &pm; 5.3 | 106.9  &pm; 2.1    |
> | antmaze-large-play-v2    | 55.9 &pm; 3.9  | 57.6 &pm; 6.6  | **58.2 &pm; 4.3**     |
> | antmaze-large-diverse-v2 | 54.0 &pm; 4.8  | 66.6 &pm; 3.1  |**68.3 &pm; 3.1**     |
>
>
> The comparison between O-DICE and AILOT+O-DICE is not quite correct as they solve different problems (offline RL and imitation learning, respectively). But it should be noted that results on hard antmaze-large-play and antmaze-large-diverse tasks outperform those of O-DICE. This is of no surprise since temporal grounding is lacking in the O-DICE method, where only distribution matching with behavior dataset is performed.
>
> ***Comprehensive Sensitivity Analysis for Cost Function and Hyperparameters:***
>
> Hyper-parameters for the main algorithm in squashing function for rewards were chosen to be consistent with those in OTR for ease of comparison. In our experiments we found that values of a < 1, a > 5 drop performance significantly and this is due to an inappropriate range of resulting rewards. We conducted experiments with varying parameters on hard antmaze tasks. Overall, across different environments the drop is small.
>
>
> | Dataset               | AILOT (a = 2, tau=1) | AILOT(a=3, tau=1) | AILOT(a=3, tau=2) |
> | --------------------- | -------------------- | ----------------- | ----------------- |
> | antmaze-large-diverse | 63.4 +-2.1           | 63.1 +- 2.0       | 64.6 +- 3.4       |
> | antmaze-large-play    | 56.3 +- 4.6          | 56.1 +- 3.7       | 57.3 +- 4.1       |

---

> > ### Author Response · Authors · 2024-11-28
> >
> > Given the discussion period is about to end, please kindly let us know if you are satisfied with the revised paper and our answers. If our answers and the extra experiments have met your expectations, we would greatly appreciate if you could revise the score. Thank you.

---

> > > ### Comment · Reviewer_LCQn · 2024-12-02
> > >
> > > Thank you for your responses. I will keep my previous score.

---

### Official Review · Reviewer_ANYp · 2024-11-03

**Soundness:** 3
**Presentation:** 3
**Contribution:** 3
**Rating:** 6
**Confidence:** 3

**Summary:**

This paper focuses on practical offline reinforcement learning tasks with only expert observations, avoiding the requirements for expert actions and reward labels. Specifically, this paper proposed AILOT (Aligned Imitation Learning via Optimal Transport), which defines the intrinsic rewards using optimal transport distance between the intention representations of the expert’s and agent’s trajectories. Through dense reward relabeling, AILOT outperforms state-of-the-art offline imitation learning methods and improves other offline reinforcement learning methods on the D4RL benchmarks.

**Strengths:**

1.	The proposed AILOT method eliminates the requirements of expert rewards and actions. Instead of performing Optimal Transport matching, AILOT maps the initial state space to the space of intentions and aligns the intents of the agent with those of the expert via Optimal Transport.  This approach involves several steps: 1) training general-purpose value functions from the expert dataset to learn the metric-aware representations; 2) solving the Optimal Transport alignment to obtain the coupling matrix; 3) reward labeling for the expert observations using the coupling matrix; and 4) training RL using the expert dataset with labeled rewards to obtain the final policy.
2.	The intent differences between the k-step state representations have a linear dependence on the step count. This near-monotone function reflects the global geometric dependencies between states in the expert dataset. This good property is important for defining the cost function of Optimal Transport alignment learning.
3.	The dense reward from AILOT can also boost the performance of other offline reinforcement learning methods. The performances of offline imitation learning and offline reinforcement learning have been demonstrated in the extensive experiments on D4RL benchmarks.

**Weaknesses:**

1.	AILOT is built on top of OTR, following the idea of performing reward relabeling through optimal transport. The most interesting part of AILOT is to perform Optimal Transport alignment in the space of intention instead of the original state space. However, the intention learning method is an existing work called ICVF, which limits the novelty.
2.	Optimal Transport introduces additional runtime overhead compared to the offline RL algorithms, with the benefits of reward labeling.

**Questions:**

1.	In the experiments, AILOT is applied with Implicit Q-Learning (IQL) because it is a simple and robust offline RL algorithm. Is there any special reason or motivation for using IQL here, and will AILOT also perform well with any other offline RL algorithm?

---

> ### Author Response · Authors · 2024-11-25
>
> Thank you very much for your time and detailed consideration of our work and for highlighting the key strengths of the paper!
>
> ***Optimal Transport introduces additional runtime overhead compared to the offline RL:***
>
> The Runtime Section 5.2 demonstrates that the optimal transport incurs minimal computational overhead - no more than ~10 minutes beyond the standard offline RL processing time. Importantly, because these calculations take place in a fixed latent space of intents, the added computation time remains stable across different tasks, *independent of their state dimensionality*.
>
> ***The choice of IQL as the base offline RL algorithm:***
>
> First of all as you also noted IQL is known for its simplicity and robust performance across different tasks. But the specific reason for this choice is that OTR (the main baseline) also includes IQL and it makes the comparison with OTR more meaningful because we can take the results from their article. Also by construction AILOT's reward relabeling approach is RL-algorithm-agnostic. So the performance improvements over baselines suggest the benefits come from better reward signals rather than the choice of base RL algorithm.
> AILOT can be combined with other RL methods like Diffusion-QL (see Table 8 in Appendix) or  more recent O-DICE (L. Mao et al. ICLR'2024).
>
> To substantiate this, we have combined AILOT with the recent state-of-the-art RL method method O-DICE (L. Mao et al. ICLR'2024) and performed the corresponding experiments. The results are summarized in the table below.
>
> | Dataset               | O-DICE     | AILOT + IQL      | AILOT + O-DICE |
> | --------------------- | ---------- | ---------- | -------------- |
> | halfcheetah-medium-v2         | 47.4 &pm; 0.2  | 47.7 &pm; 0.2 | **49.5 &pm; 0.4**     |
> | halfcheetah-medium-replay-v2       | 44.0  &pm; 0.3 | 42.4 &pm; 0.2  | **46.2 &pm; 0.6**    |
> | halfcheetah-medium-expert-v2       | 93.2 &pm; 0.6  | 92.4 &pm; 1.5  | **93.6 &pm; 1.8**   |
> | hopper-medium-v2              | **86.1 &pm; 4.0**  | 82.2 &pm; 5.6  | 85.5 &pm; 3.7      |
> | hopper-medium-replay-v2             | **99.9 &pm; 2.7**  | 98.7 &pm; 0.4  | 99.1  &pm; 0.2     |
> | hopper-medium-expert-v2           | **110.8 &pm; 0.6** | 103.4 &pm; 5.3 | 106.9  &pm; 2.1    |
> | antmaze-large-play-v2    | 55.9 &pm; 3.9  | 57.6 &pm; 6.6  | **58.2 &pm; 4.3**     |
> | antmaze-large-diverse-v2 | 54.0 &pm; 4.8  | 66.6 &pm; 3.1  |**68.3 &pm; 3.1**     |

---

> > ### Comment · Reviewer_ANYp · 2024-11-26
> >
> > Thank you for your responses. I will keep my previous score.

---

### Official Review · Reviewer_2N2x · 2024-11-04

**Soundness:** 3
**Presentation:** 1
**Contribution:** 2
**Rating:** 6
**Confidence:** 3

**Summary:**

This paper considers learning from offline data in settings where reward may be difficult to specify, but one (or multiple) expert trajectories demonstrating the behavior may be found. The general idea is to assign rewards within a trajectory based on an optimal transport distance between trajectories in the offline data, and this optimal trajectory. The primary innovation is to use a dynamical distance (ICVF) to parameterize a more semantically meaningful cost function for the optimal transport problem. The evaluation demonstrates improvement over prior approaches in this problem setting on all state-based D4RL tasks (including locomotion, antmaze, and adroit).

**Strengths:**

The problem setting is topical, and the method is simple and well-motivated -- Figure 1 in particular illustrates clearly the benefit of parameterizing distances in a latent space instead of the raw state space (or equivalent).

The experiments clearly demonstrate performance improvement over prior approaches (both based on optimal transport, or other imitation learning) in the D4RL suite. Admittedly, these tasks are relatively toy and now saturated, but even so, the results seem convincing.

The related work throughout the paper (in intro, related work, and method section) contextualizes the contributions of this paper well.

The appendix (and experimental section) thoroughly describes the comparisons and the benchmark setting.

**Weaknesses:**

I found the writing in the paper to be difficult to comprehend at many parts, making it difficult to understand the method exactly and what the exact contributions are relative to prior work in this space (e.g. Luo et al).

For instance, the introduction barely touches on the method being proposed, instead discussing in great detail the motivation for IL methods, for optimal transport, etc. This makes it difficult to understand and contextualize the specific contributions of the method being proposed in the paper.

The paper is most closely related to Luo et al, 2023 (OTR), but within the method section, does not distinguish between what ideas come from Luo et al, and which are newly introduced in this paper. For readers who may not be familiar with this prior work, this can lead to misattribution of ideas. It would be useful (whether in the related work, background, or method) to more clearly lay out what is done in Luo et al, and what new ideas are being considered.

The novelty of the idea (to my understanding) over Luo et al is relatively low -- this, in itself, is not a bad thing. However, given the simplicity of the idea, it would have been nice to see more thorough ablations and analyses to understand how e.g different dynamical distances perform, what types of data this is most helpful with, the importance of both components of the cost function. Another axis that could improve the thoroughness of the paper is to evaluate on more challenging domains beyond where standard cost metrics succeed (for example, in image-based domains). One other possible avenue of improvement here may be to thoroughly investigate what the actual computed rewards look like between this method and prior work.

As it stands right now, while the method demonstrates mild improvements on D4RL, the paper could be much improved by expanding the analysis on the axes why the learned representation is much more useful, or by testing on a more difficult suite of tasks.

**Questions:**

1. Why is there minimal benefit to scaling the number of expert trajectories? How well would this method handle using expert trajectories that take different behaviors to solve the same problem (for example, the Push-T task from Diffusion Policy)

2. Could you explain better what the two different components of the cost function are doing? The text didn't well-motivate why these were chosen in this way.

3.  How sensitive is the method to `k`?

4. Would be nice to expand the discussion about how this method handles sub-optimal / orthogonal data compared to traditional offline algorithms -- Can this method "stitch" trajectories together?

---

> ### Author Response · Authors · 2024-11-14
>
> Thank you very much for your thorough and insightful review. Your questions help to clarify important aspects of our work. I'll address each point carefully below.
>
> ***Differences between this Approach and OTR***
>
> You are correct that the main differences are: Our focus on intent-based representations rather than raw state-action pairs.
> The modified structure of matrix C (eq. 10) with k=2 instead of k=1.
>
> Supporting evidence: See ablation studies in Tables 1,2,3,6,7
>
> We also investigated the properties of the intent distance metric itself. We proved theoretically (Proposition 1) and experimentally (Figure 3) that the chosen metric corresponds to the stated hypothesis. Such results are not presented in previous works of OTR (Luo et al., 2023) and ICVF (D. Ghosh et al., 2023). This is a new property and a new application of intents, which we proposed in this paper.
>
> ***Minimal Benefit from Scaling Expert Trajectories***
> According to the algorithm 1 (step 15) we take the minimum over expert trajectories. That is, in particular, it searches for the closest expert trajectory to the agent's trajectory. Additional similar expert trajectories may provide redundant information without adding new insights. If there are expert trajectories that take different behaviors to solve the same problem and the policy is stochastic (like in IQL) then AILOT will imitate all these different behaviors.
>
> ***Cost Function Components***
> First Term: Aligns current states in intent space.
> Second Term: Ensures temporal consistency via future state alignment.
> Together they enforce both spatial and temporal alignment.
>
> Reference: Lines 262-269 in paper for detailed motivation
>
> ***Sensitivity to Parameter k***
> Both OTR and AILOT are not sensitive to k. But setting k=2 gives a small improvement. Smaller k=1 may not capture enough temporal context, while larger k>2 values makes alignment harder.
>
> Evidence: See Table 7 for experimental results.
>
> ***Trajectory Stitching Capability***
> Yes. According to the algorithm 1 (step 15) it can stitch rewards from different expert trajectories.

---

> > ### Author Response · Authors · 2024-11-28
> >
> > Given the discussion period is about to end, we wanted to inquire if we had addressed the concerns. Please kindly let us know if we can do anything else to deserve the extra points in our score. Thank you.

---

### Author Response · Authors · 2024-11-27

Dear Reviewers,

Thank you for your valuable feedback that has greatly improved our manuscript.

We have carefully considered and addressed all of your comments, making the necessary revisions and running **14 additional experiments** during the rebuttal. In particular, we conducted experiments in combination with another recent RL method O-DICE (L. Mao et al. ICLR 2024), confirming that AILOT is not tied to a specific RL algorithm and works well with any of them. We hope our responses meet your expectations.

It is also worth emphasizing that we not only proposed a new distance metric between states, which performed well in experiments, but also studied its properties, giving an answer to why it is better than the conventional Euclidean metric. This drastically distinguishes our method from the OTR (Luo et al, 2023) baseline.

To summarize, we showcased that AILOT is the state-of-the-art Imitation Learning algorithm. Specifically, **we outperformed 8 powerful SOTA methods in 32 benchmarked dataset tasks**. Thanks to OT, our agent managed to do it merely by observing and imitating the expert (no labels, no ground truth rewards). Given the long history of RL and the elegant proposal to align intentions, we long for an opportunity to present these results and ideas to the ICLR community.

If there are any remaining concerns or suggestions, we are available for prompt responses to any queries.

Thank you once again for your time and all the insightful feedback!

Best regards,

Authors of Submission 7886

---

### Meta-Review · Area_Chair_Cm6B · 2024-12-21

**Metareview:**

This paper proposes a method for the offline RL setting where rewards are difficult to specify but one (or multiple) expert trajectories demonstrating the behavior may be found. The proposed method computes rewards by compares the optimal transport distance (computed via ICVF) between a new trajectory and the expert trajectory. The proposed method outperforms state-of-the-art offline imitation learning methods and improves other offline reinforcement learning methods on the D4RL benchmarks, especially in sparse-reward environments.

Reviewers appreciated the strong empirical performance, the strong motivation (esp. Fig 1) and the discussion of prior work. They also noted appreciated the thorough experimental details.

One repeated concern was the relationship with Luo et al (OTR), and urged the authors to revise the paper to include a more detailed discussion of the relationship with this prior work to avoid misleading readers into misattributing some of the ideas discussed in this paper. Given the similarity with Luo et al, the reviewers were looking for more thorough experiments (e.g., when and why does using the learned metric help). Reviewers also requested an additional baseline (O-DICE) and more details about the training setup.

Overall, I feel like this has the makings of a strong paper, but the paper needs additional revisions/analysis to address the concerns about similarity with Luo et al.

**Additional Comments On Reviewer Discussion:**

During the rebuttal, the authors ran several new experiments (e.g., adding the requested O-DICE baseline), clarified some of the differences from prior work, the computational cost of the proposed method, and the construction of the cost matrix.

These revisions and experiments address some of the reviewer concerns, but don't seem to get at the most prevalent concern (relationship with Luo et al). Indeed, when I read the introduction of the revised paper, the similarity/difference from Luo et al is not immediately apparent.

---

### Decision · Program_Chairs · 2025-01-22

Reject